# Transferring Know-How for an Autonomous Camera Robotic Assistant

**Irene Rivas-Blanco** *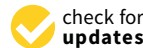, **Carlos J. Perez-del-Pulgar** 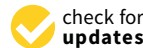, **Carmen López-Casado** 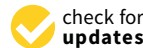, **Enrique Bauzano** 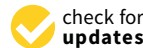 and **Víctor F. Muñoz**

Department of Systems Engineering and Automation, Universidad de Málaga, Andalucía Tech, 29071 Málaga, Spain; carlosperez@uma.es (C.J.P.-d.-P.); mclopezc@uma.es (C.L.-C.); ebauzano@uma.es (E.B.); vfmm@uma.es (V.F.M.)

*   Correspondence: irivas@uma.es; Tel.: +34-952-137-204

**Abstract:** Robotic platforms are taking their place in the operating room because they provide more stability and accuracy during surgery. Although most of these platforms are teleoperated, a lot of research is currently being carried out to design collaborative platforms. The objective is to reduce the surgeon workload through the automation of secondary or auxiliary tasks, which would benefit both surgeons and patients by facilitating the surgery and reducing the operation time. One of the most important secondary tasks is the endoscopic camera guidance, whose automation would allow the surgeon to be concentrated on handling the surgical instruments. This paper proposes a novel autonomous camera guidance approach for laparoscopic surgery. It is based on learning from demonstration (LfD), which has demonstrated its feasibility to transfer knowledge from humans to robots by means of multiple expert showings. The proposed approach has been validated using an experimental surgical robotic platform to perform peg transferring, a typical task that is used to train human skills in laparoscopic surgery. The results show that camera guidance can be easily trained by a surgeon for a particular task. Later, it can be autonomously reproduced in a similar way to one carried out by a human. Therefore, the results demonstrate that the use of learning from demonstration is a suitable method to perform autonomous camera guidance in collaborative surgical robotic platforms.

**Keywords:** surgical robotics; human–machine interaction; autonomous guidance

## 1. Introduction

In the last decades, surgical robotics has spread to the operating rooms as a daily reality. The Da Vinci surgical system (Intuitive Surgical, Inc.), the most used commercial surgical robot, has distributed more than 4000 units in hospitals around the world that have been used to perform more than five million procedures. This platform is composed of a slave side that replicates the motions, and a surgeon who performs in a master console [1]. Although this kind of robot enhances the surgeon's abilities, providing more stability and accuracy to the surgical instruments, their assistance is limited to imitating the movements performed on the master console. However, all the cognitive load lies on the surgeon, who has to perform every single motion of the endoscopic tools. The implementation of collaborative strategies to perform autonomous auxiliary tasks would reduce the surgeon's workload during the interventions, letting she or he concentrate on the important surgical task.

Camera guidance is a particularly interesting task to be automated, as it is a relatively simple but crucial task that may significantly help the surgeons. Most robotic surgeries are performed by a single surgeon who controls both the instruments and the camera, switching the control between them through a pedal interface. Omote et al. [2] addressed the use of a self-guided robotic laparoscopic

camera based on a color tracking method to follow the instruments. In this work, they compared this method with human camera control and demonstrated that autonomous camera guidance slightly reduces the surgery time, and the camera corrections and lens cleaning frequency were drastically improved. Therefore, this approach helps the surgeon to be more concentrated on the surgery.

Current camera guidance strategies can be divided into three approaches; direct control interfaces, instruments tracking, and cognitive approaches. Direct control interfaces, such as voice commands [3], head movements [4], or gaze-contingent camera control [5], have succeeded in substituting medical staff for moving the camera. However, these methods require very specific instructions and extraneous devices that may distract the surgeon. On the other hand, camera guidance through instrument tracking gives the robot more autonomy and does not require continuous surgeon supervision [6–8]. The earliest works employed color markers to identify the tip of the tools, but the current trend is to use deep learning techniques to identify the tools in raw endoscopic images [9–11]. Reference [12] improves traditional tracking methods with the long-term prediction of the surgical tools motion using Markov chains.

The main problem of these strategies is that they follow simple and rigid rules, such as tracking only one tool, or the middle point of both tools [13], but they do not consider other important factors that affect the behavior of the camera such as the knowledge of what task the surgeon is performing at a given time or particular camera view preferences.

Hence, cognitive approaches emerge to provide more flexibility to the camera behavior, making the guidance strategy dependent on the current state of the task. Reference [14] proposes a human–robot interaction architecture that sets a particular camera view depending on the surgical stage. The camera view may be tracking the instruments or pointing at a particular anatomical structure. In our previous work [15], we propose a cognitive architecture based on a long-term memory that stores the robot's knowledge to provide the best camera view for each stage of the task. This work also included an episodic memory that takes into account particular preferences of different users. This work was improved in [16] with a navigation strategy that merges the advantages of a reactive instrument tracking with a proactive control based on a predefined camera behavior for each task stage. A reinforcement learning algorithm was used to learn the weight of each kind of control to the global behavior of the camera. This work revealed that this autonomous navigation of the camera improved the surgeon performance and did not require the interaction of the surgeon. However, this strategy requires an exhaustive hand-crafted model of the control strategy.

To solve this issue, learning from demonstration (LfD) arises as a natural means to transfer human knowledge to a robot that avoids the conventional manual programming. Essentially, a robot observes how a human performs a task (i.e., the demonstration) and then it autonomously reproduces the human behavior to complete the same task (i.e., the imitation). This approach is used in a wide variety of applications like rehabilitation and assistive robots [17], motion planning [18], intelligent autopilot systems [19], learning and reproduction gestures [20], or haptic guidance assistance for shared control applications [21,22]. In the field of surgical robotics, LfD has been used for different purposes. Reiley et al. [23] proposed the use of this method to train and reproduce robot trajectories from previous expert demonstrations, which were obtained using the Da Vinci surgical system. These trajectories were used to evaluate different surgeon skill levels (expert, intermediate and novice). On the other hand, van den Berg et al. [24] proposed the use of LfD to allow surgical robotic assistants to execute specific tasks with superhuman performance in terms of speed and smoothness. Using this approach, the Berkeley Surgical Robot was trained to tie a knot in a thread around a ring following a three-stage procedure. The results of this experiment demonstrated that the robot was able to successfully execute this task up to 7x faster than the demonstration. Recently, Chen et al. [25] propose the use of LfD combined with reinforcement learning methods to learn the inverse kinematics of a flexible manipulator from human demonstrations. Two surgical tasks were carried out to demonstrate the effectiveness of the proposed method.

This paper explores the use of LfD to guide the camera during laparoscopic surgery. A new approach to transfer human know-how from previous demonstrations is defined. It uses Gaussian

mixture models (GMM) to generate a model of the task, which is later used to generate the camera motions by means of Gaussian mixture regression (GMR). This approach has been experimentally validated through a surgical robotic platform that is composed of three manipulators, which holds two instruments and the endoscopic camera. This platform has been used to train different behaviors of the camera during a peg transferring task, which is commonly used to train surgeons skill. The information provided by the training was used to create a GMM for this task. Later on, the same robotic platform was used to reproduce the task and provide the autonomous camera guidance from the GMM. Training and reproduction were evaluated in order to validate the proposed approach to be applied in surgical robotics.

This paper is organized as follows. Section 2 describes the autonomous camera guidance method based on a LfD approach. The experiments performed to validate the proposal are detailed in Section 3, and Section 4 presents the discussion and the future work.

## 2. Autonomous Camera Guidance

Figure 1 shows the general scenario of an abdominal laparoscopic surgery, with two surgical instruments and the camera, which tip positions are defined as $\vec{p_1} = (p_{1x}, p_{1y}, p_{1z})$, $\vec{p_2} = (p_{2x}, p_{2y}, p_{2z})$, and $\vec{c} = (c_x, c_y, c_z)$, respectively. The idea of the guidance approach is to teach the system how the camera moves depending on the surgical instrument positions at a given time, as they are the main reference to establish the viewpoint. Thus, we have to set a common reference frame to relate the instruments and the camera measures, which is represented as $\{O\}$ in the figure. The most natural choice is to set a global frame associated with an important location within the particular task for which the system is going to be trained, e.g., if we are training the system to move the camera in a cholecystectomy, then a natural reference would be the gallbladder, but if the task is a kidney transplantation then it is more reasonable to take the kidney as the reference. Hence, the global frame $\{O\}$ was chosen for each particular task, and it was set in an initial calibration process at the beginning of each set of demonstrations and reproductions of the task.

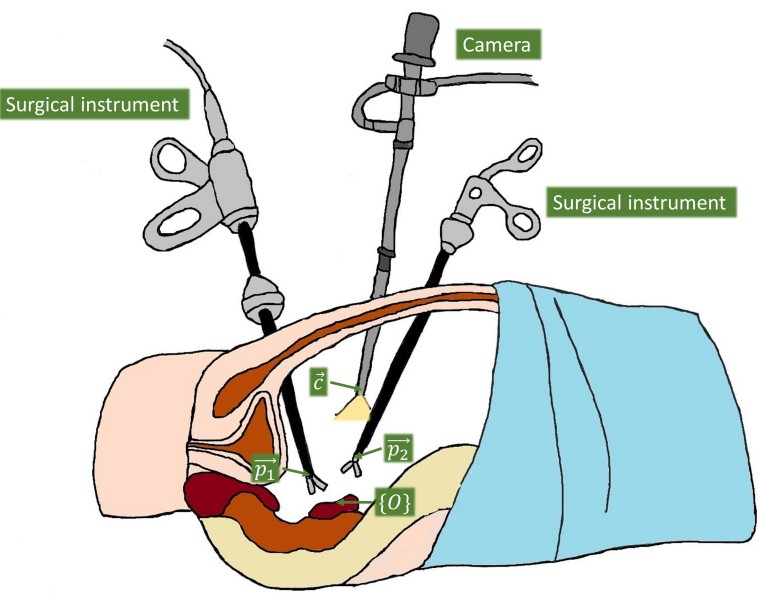

**Figure 1.** General scenario of an abdominal laparoscopic surgery.

The autonomous camera guidance method proposed in this paper followed the LfD approach shown in Figure 2, which was based on two stages: the first one, the off-line stage, created the robot knowledge base from human expert demonstrations; and the second one, the on-line stage, used that knowledge to generate the camera motion. During the off-line stage, an expert surgeon performed a

set of demonstrations of the camera motion for a particular task, and the system stored the camera position and its relation to the current position of the surgical instruments, i.e., the camera position $\overrightarrow{c}$ for each tuple of instrument positions $(\overrightarrow{p_1}, \overrightarrow{p_2})$. The demonstrations can be carried out with the surgical robotic platform used in the experiments, or using other devices, such as joysticks. Moreover, conventional surgical tools could also be employed, using a tracking position sensor as it was done in [26] for surgical manoeuvre recognition purposes. The objective of this process, called know-how transferring, was to create a knowledge base that stores the behavioral model of the camera ($\rho$). Then, during the on-line stage, the motion generation module took the previously trained model $\rho$ and the current position of the surgical instruments, $\overrightarrow{p_1}$ and $\overrightarrow{p_2}$, to update the camera location, $\overline{c}$. To ensure the safety of the patient in a real surgery, the system must include a human–machine interface (HMI) that allows the surgeon to take control of the camera in case of an undesirable motion, and a supervisor system that guarantees that the camera moves inside a safety region.

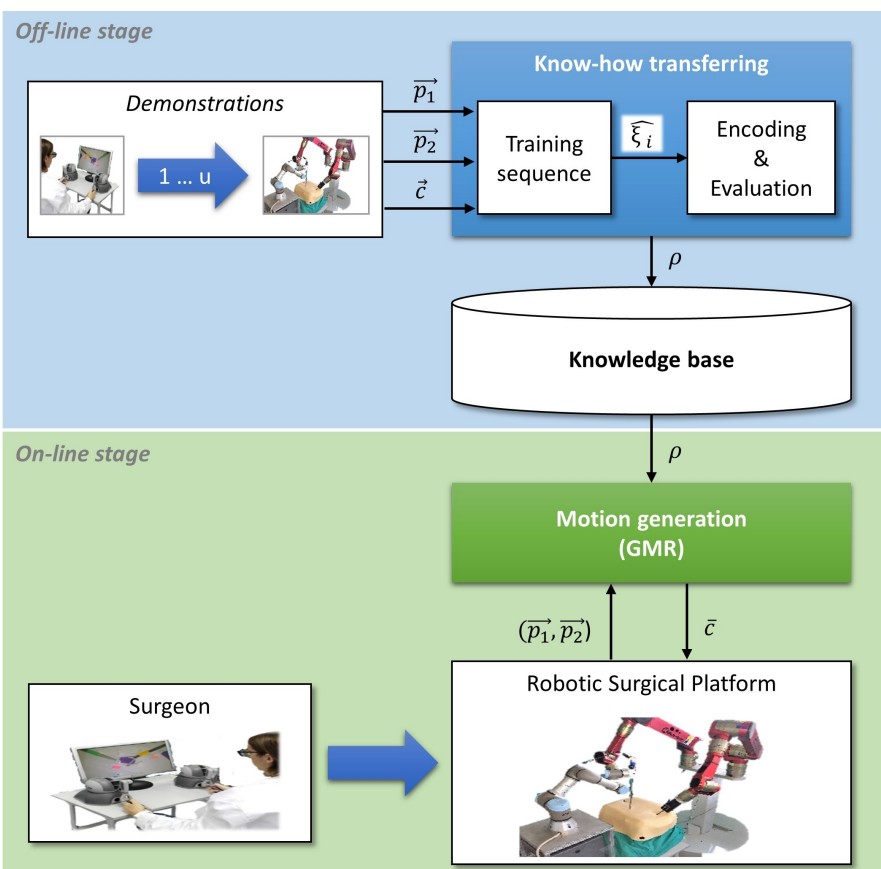

**Figure 2.** Learning for demonstration approach for autonomous camera guidance.

## 2.1. Know-How Transferring

Know-how transferring is the process through which an expert surgeon demonstrates the camera motion for a particular surgical task. Starting from a data tuple that contains the position of the instruments and camera, which can be defined as follows,

$$\xi = (p_{1x}, p_{1y}, p_{1z}, p_{2x}, p_{2y}, p_{2z}, c_x, c_y, c_z) \tag{1}$$

each demonstration *i* generated a training sequence of *k* data tuples as:

$$\hat{\xi}_i(k) = \xi(1), ..., \xi(k), 1 \le i \le u. \tag{2}$$

Once the demonstrations were performed, the generated training sequences were encoded into a GMM $\rho$. A GMM is a probabilistic model that is able to include the training sequences in a set of $N$ Gaussians, whose mixture covers the training sequences. So, a GMM can be defined as:

$$\rho = \{\pi_n, \mu_n, \Sigma_n\}_{n=1}^{N}, \tag{3}$$

where a Gaussian $n$ is represented by $\pi_n$, which is the weight of each $n$ Gaussian on the GMM, and $\mu_n$ and $\Sigma_n$ are the mean and covariance matrix of the Gaussian, respectively.

The training of a GMM was carried out by means of the expectation-maximization (EM) algorithm [27]. The EM is an iterative algorithm that estimates the values of the $N$ Gaussians of the training sequences, maximizing the likelihood of the training sequences belonging to the encoded GMM. The EM algorithm can be defined as follows, where the inputs are all the $u$ training sequences and the number of Gaussians $N$:

$$\rho = \mathrm{EM}(\hat{\xi}_{1...u}, N). \tag{4}$$

Once the model has been trained, it can be evaluated through the Bayesian information criterion (BIC) as follows:

$$\mathrm{BIC} = -L((k)) + \frac{n_p}{2}\log(k) \tag{5}$$

where $L$ is defined as:

$$L(\hat{\xi}(k)) = \sum_{j=1}^{k} \log\left(P\left(\xi\left(j\right)\right)\right) \tag{6}$$

with:

$$P(\xi) = \sum_{n=1}^{N} P(n)P(\xi|n) \tag{7}$$

In this equations, $P(n)$ is the prior probability and $P(\xi|n)$ is the conditional probability density function, both defined as:

$$P(n) = \pi_n, \tag{8}$$
$$P(\xi|n) = N(\xi, \mu_n, \Sigma_n). \tag{9}$$

Finally, in Equation (5), $n_p$ is a variable used to penalize the score taking into consideration the dimension of the tuple, $D$, and the number of Gaussians, $N$, as follows:

$$n_p = (N-1) + N(D + \tfrac{1}{2}D(D+1)). \tag{10}$$

This method provided a score that was used to choose the best number of Gaussians $N$. The lower the score, the better the model fitness was [28].

*2.2. Motion Generation*

Once the model has been trained within $\rho$ (Equation (3)) and evaluated, the camera motion could be extracted through GMR [27]. For this purpose, the parameters of $\rho$ can be represented as:

$$\mu_n = \begin{bmatrix} \mu_n^p \\ \mu_n^c \end{bmatrix}, \quad \Sigma_n = \begin{bmatrix} \Sigma_n^p & \Sigma_n^{pc} \\ \Sigma_n^{cp} & \Sigma_n^c \end{bmatrix}. \tag{11}$$

These parameters were used to obtain the camera position using GMR as stated in Equation (12), where $\bar{c} = (c_x, c_y, c_z)$ is the camera position generated for a particular position of the instruments $\vec{p_1} = (p_{1x}, p_{1y}, p_{1z})$ and $\vec{p_2} = (p_{2x}, p_{2y}, p_{2z})$:

$$\bar{c} = \sum_{n=1}^{N} P(n|(p_{1x}, p_{1y}, p_{1z}, p_{2x}, p_{2y}, p_{2z}) \left[ \mu_n^c + \frac{\Sigma_n^{pc}}{\Sigma_n^p}((p_{1x}, p_{1y}, p_{1z}, p_{2x}, p_{2y}, p_{2z}) + \mu_n^p) \right], \qquad (12)$$

where

$$P(n|(p_{1x}, p_{1y}, p_{1z}, p_{2x}, p_{2y}, p_{2z})) = \frac{P(n)P((p_{1x}, p_{1y}, p_{1z}, p_{2x}, p_{2y}, p_{2z})|n)}{\sum_{j=1}^{N} P(j)P((p_{1x}, p_{1y}, p_{1z}, p_{2x}, p_{2y}, p_{2z})|j)}. \qquad (13)$$

These equations can be used every sample time to generate the camera position $\bar{c}$ and send it to the surgical robotic platform in order to autonomously move the camera to the corresponding position.

## 3. Experimental Results

The autonomous camera guidance method, proposed in the previous section, has been validated through two experiments, each one for each stage of training and task reproduction. The training stage showed how a surgical task could be trained using GMM, and the second one, task reproduction, showed the camera behavior based on the previously learned knowledge to perform autonomous camera motions. These experiments have been carried out using the surgical robotic platform described below.

### 3.1. Experimental Scenario

Figure 3 shows the experimental scenario used to perform the validation of this work. The surgical robotic platform was composed of three arms; two of them that make up the CISOBOT platform, in charge of the teleoperation of the surgical instruments, and a UR3 robot, from Universal Robots, with a commercial 2D endoscope attached at its end-effector. The CISOBOT is an experimental platform developed at the University of Malaga, which is composed of two customized six degrees-of-freedom manipulators with a laparoscopic grasper tool attached at their end-effectors [29]. These robots are teleoperated using the master console that is shown in Figure 3, which is composed of two haptic devices and a monitor that displays the image of the camera. The surgeon used two commercial haptic devices, without force feedback, to teleoperate the robots (Phantom Omni, Sensable Technologies), and during the training stage, an additional haptic was used to move the camera. The master console was placed in the same room as the robotic platform, but the surgeon did not have direct vision of the task, so the only visual feedback he/she had is the image of the camera.

The instruments were inserted in an abdomen simulator with the experimental board shown in Figure 4 inside. It was a commercial pick-and-place pegboard developed by Medical Simulator used to train basic laparoscopic skills. This pegboard was 125 mm × 125 mm, and it had six cylindrical rubber rings, which the user (usually a novice surgeon) had to transfer from one peg to another. This was one of the five tasks described in the SAGES manual of skills in laparoscopic surgery [30] to measure technical skills during basic laparoscopic surgical maneuvers. The main purpose of this task was to exercise depth perception in a 2D environment, where the only visual feedback we had was the image of the camera. Thus, this was a suitable task to evaluate the autonomous camera guidance method proposed in this work, as the camera view was crucial for the successful performance of the task.

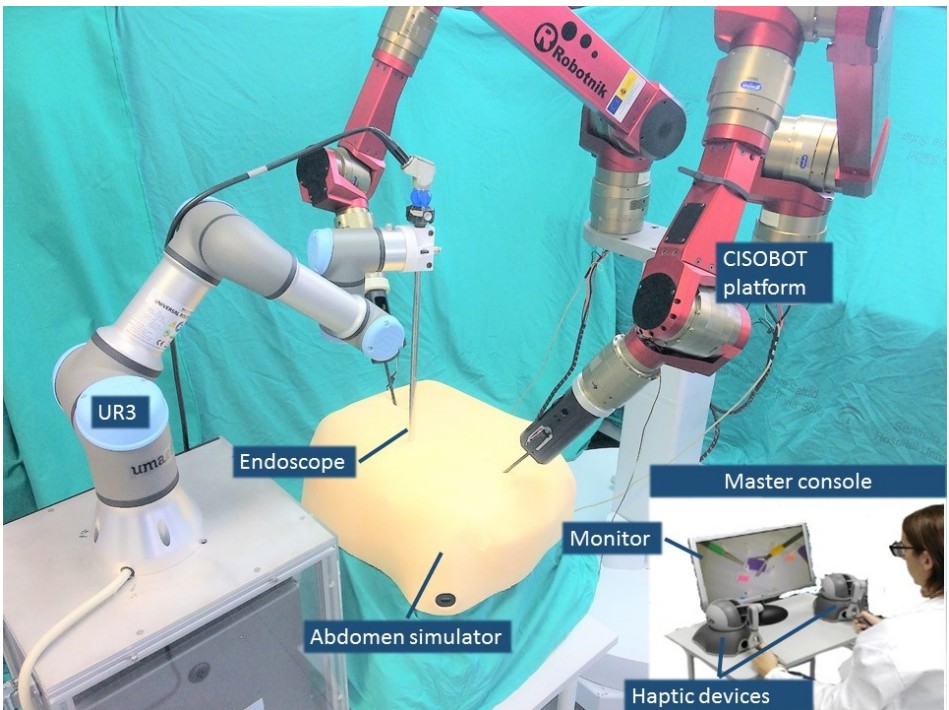

**Figure 3.** Experimental surgical robotic platform. It was composed of a teleoperation master console, an endoscope that was handled by a UR3 robot, and the CISOBOT platform.

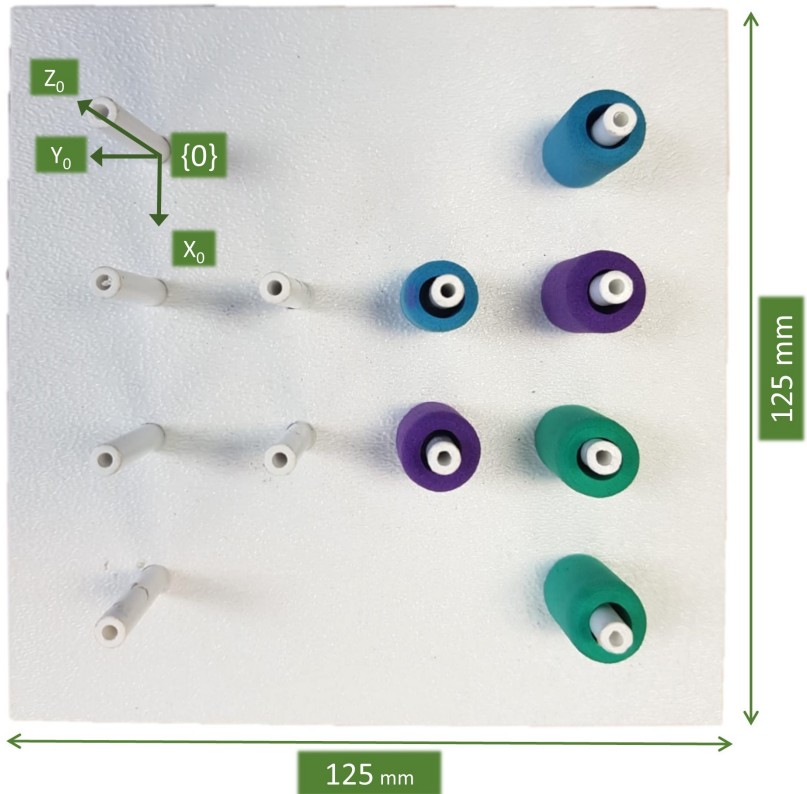

**Figure 4.** Experimental board. It had 12 pegs and the objective was to move cylindrical rubber rings from one peg to another.

For this task, the global frame $\{O\}$ was set at the top left peg of the pegboard, as shown in Figure 4. The global frame calibration process comprised of two steps: first, the user touched the top left peg

with the tip of the right surgical tool to establish the relation between the pegboard and the CISOBOT; and second, the user touched the tip of the camera with the same tool to relate the CISOBOT and the UR3 robot. Afterward, these relations were stored in the system, and all the data measured during the performance of the experiments was transformed to the global frame $\{O\}$.

### 3.2. Data Acquisition

The implementation of the method for autonomous camera guidance in laparoscopic surgery follows the software/hardware architecture shown in Figure 5. It was based on the open-source framework ROS [31], which allows easy communication between the different components of the system. The two main nodes of the software architecture are the *tool teleoperation node* and the *UR3 robot node*, both running at 125 Hz. The CISOBOT platform was controlled by real-time hardware (NI-PXI, http://www.ni.com/en-nz/shop/pxi.html) that provided natural teleoperation of the surgical tools. This control was integrated into a ROS node that published the position of the tools to the rest of the system. On the other hand, the camera had two ways of operating; during the off-line stage, the camera was teleoperated using ROS to communicate the Phantom Omni with the UR3 robot; and during the on-line stage, the camera followed an autonomous guidance according to the motion generated by the GMR model, which was implemented in a MATLAB environment.

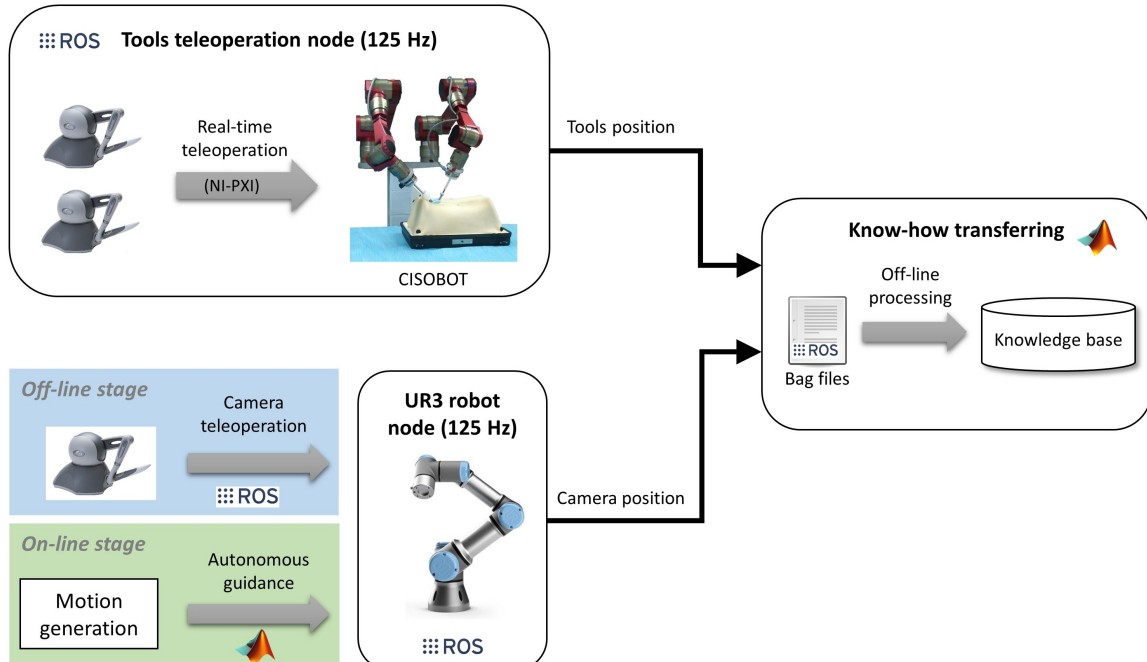

**Figure 5.** SW/HW architecture for the autonomous camera guidance method.

Finally, during the know-how transferral process, the training data was stored in a bag file format, which allowed recording synchronized data published by different ROS nodes for off-line analysis. Hence, the bag files recorded the surgical tool positions from the CISOBOT node and the camera position from the UR3 node. Then, that data was encoded in MATLAB to generate the knowledge base containing the GMM of the camera behavior.

The training and reproduction data, as well as the source code, implemented to perform this work have been published as an open repository in github (https://github.com/spaceuma/LfDCameraGuidance).

### 3.3. Training

The training stage of the autonomous camera guidance approach consisted of teaching the system how to move the camera during the pick-and-place task described above. Hence, the training needed

two people: the main surgeon that moved the tools and a surgeon assistant that handled the camera. The task consisted of picking the rings with one of the tools, transferring it to the other tool, and placing it in another peg of the pegboard. As the objective was to learn a general behavior of the camera, there was no predefined order to pick-and-place the rings. The task began with the six rings placed in the positions shown in Figure 4. Then the surgeon can freely choose both which ring to pick up and the peg to place them. When a ring falls, the surgeon can pick it again with one of the tools, unless it falls out of the workspace of the robots. When finished, the rings were placed back to the original position by hand, and the task was repeated again. In total, the training dataset contained k = 2616.34 s · 125 Hz = 327.043 tuples $\xi$, which corresponded to the duration of the training multiplied by the recording rate.

Before starting the training of the system, the surgeon and the assistant agreed on the behavior of the camera depending on the instruments relative position. This behavior had been defined so that the method was as general as possible, trying to minimize the effect of the particular preferences of different surgeons. Therefore, after this a priori conversation, the surgeon was asked to not provide additional on-line instructions to the assistant to modify the camera view. This way, the behavior of the camera followed the general guidelines independently of the surgeon's preferences. The agreed qualitative behavioral model of the camera can be divided into the following guidelines:

- Instruments tracking: in the horizontal plane (xy plane), the camera must follow the tools trying to keep always both instruments in the field of view but focusing on the active tool in case only one of them is moving. In the vertical plane (z plane), the camera should follow the tools from a certain distance that provides a suitable trade-off between field of view and zoom. Particular grades of zoom are performed by following the inward trajectories of the instruments.
- Zoom-out: in a typical laparoscopic procedure, surgeons operate in very specific areas and they need the camera to focus in that particular zone. However, sometimes they need to have a global vision of the operating area, i.e., to zoom-out the camera from the surgical tools. As this fact depends on the needs of the surgeon at a particular time, the surgeon and the assistant need a non-verbal signal to teach the system when to zoom-out the camera. For this task, the system has been trained to zoom-out when both tools make a synchronized outwards motion.

Figure 6 shows an example of the training trajectories and the GMM fitness for the xy plane with respect to the reference frame {O} for both instruments. The trajectories are represented with blue lines and the Gaussians within the GMM are represented with green ellipses. Finally, the pegboard area is represented by a grey square. As shown, the Gaussians covered the trained trajectories correctly. Comparing Figure 6a,b, it can be appreciated how the left tool covered a greater part of the left area of the pegboard, while the right reached the zone further to the right. Trajectories out of the pegboard correspond with failure situations, in which one ring has fallen out of the tools outside the pegboard limits.

Similarly, Figure 6c shows the trained camera trajectories and the learned GMM model. In this figure, the fulcrum point of the camera is represented by a red circle. This point has been chosen so that the camera view covers all the pegboard from its outer position. As the camera does not enter into the abdomen as much as the surgical tools, it does not reach as far as x and y positions as the instruments.

The number of Gaussians $N$ is an important parameter within a GMM, so its appropriate choice is critical for the system training. Indeed, a low $N$ would provide a poorly trained model and a high $N$ would increase the CPU time, which would affect the real-time constraints. To decide the best number of Gaussians, the BIC score was defined in Equation (5). Figure 7 represents the BIC score for $1 \leq N \leq 30$. As shown, the score decreases as $N$ increases, which means that the model better fits the training sequence as the number of Gaussians is increased. Therefore, the main limitation was the CPU processing time for real-time constraints. Using the proposed architecture (Figure 5), which used MATLAB Simulink Desktop Real-Time and ROS, a value $N = 20$ was reached. Higher values than this one made it impossible to be executed on the platform.

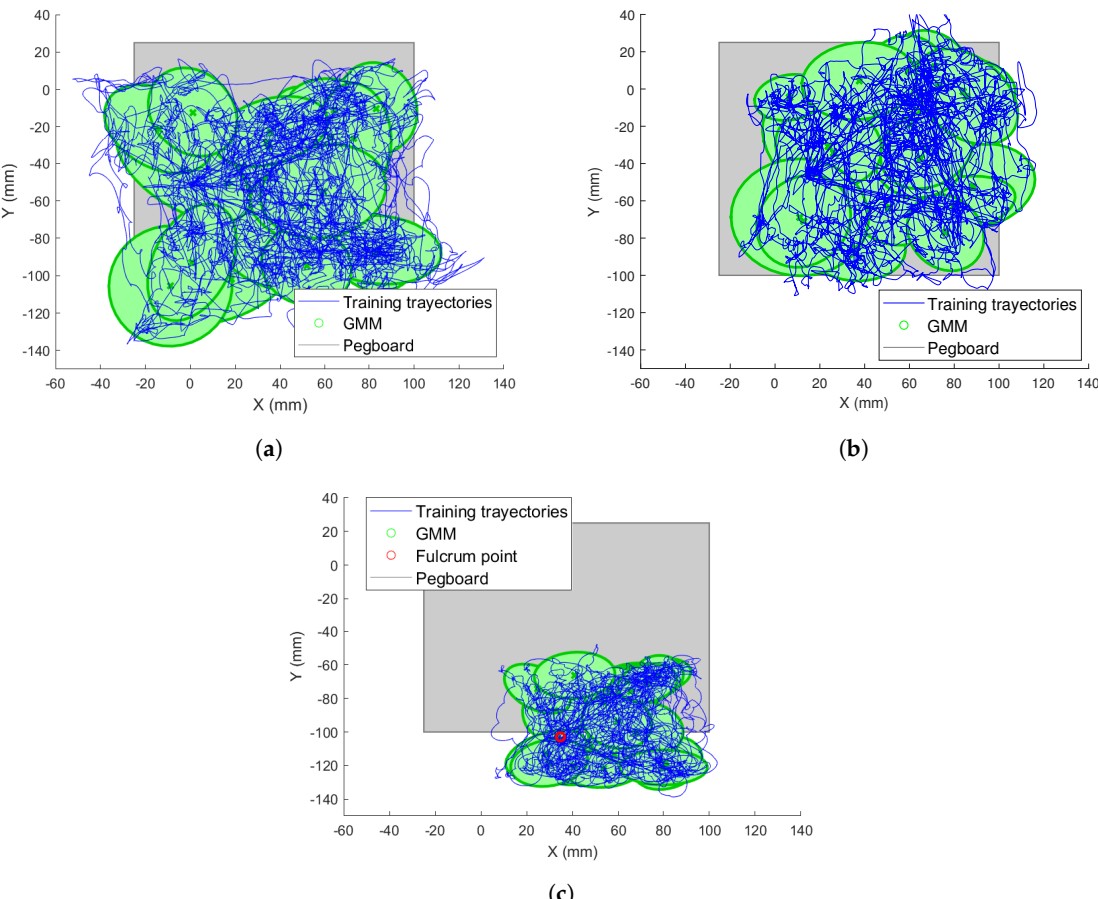

**Figure 6.** Trajectories and Gaussian mixture models (GMM) model in the xy plane during the training stage: (**a**) left tool; (**b**) right tool; and (**c**) camera.

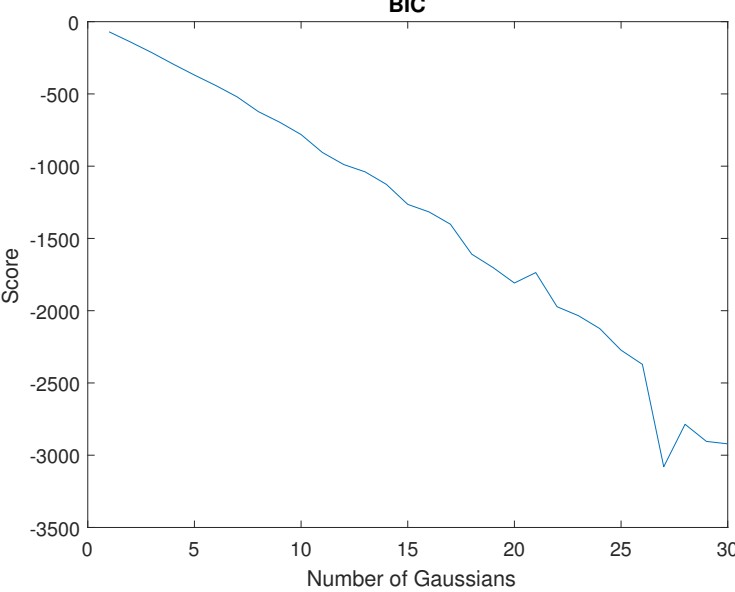

**Figure 7.** Bayesian information criterion (BIC) score for the generated GMM using the training sequence. The score is decreased (improved) as the number of gaussians are increased.

### 3.4. Task Reproduction

To validate the autonomous camera guidance method, the main surgeon has repeated the pick-and-place task described above during a period of 10 min, recording a total of 78.712 data tuples. The reproduction of the task began with the rings placed in the same positions as at the beginning of the training, i.e., as shown in Figure 4. Then, during the task reproduction, the surgeon can freely choose which ring to pick and the location to place it. This way, it is assured that the training and the reproduction tasks were not performed in the same order. During this experiment, the camera robot followed autonomously the previously trained behavior. Figure 8 shows the tools and the camera trajectories during this validation experiment. As it can be seen, the tools moved within the pegboard area, and the camera moved according to the trained positions of Figure 6c. During the whole task reproduction, the system provided a camera view that allowed the surgeon to perform the task without the necessity to manually change the camera position. A video showing a representational part of this task reproduction can be found in the Supplementary Materials.

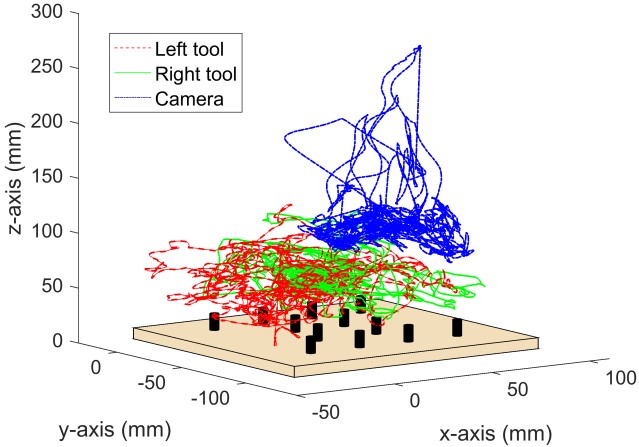

**Figure 8.** Tools and camera trajectories during the pick-and-place task reproduction.

Figure 9 shows a comparison of the instrument tracking behavior during the training stage of the system and the task reproduction. The results are divided in the trajectories in the horizontal xy plane (Figure 9a) and in the vertical z plane (Figure 9b). In the horizontal plane, it can be seen that the camera moved within the trained area. This figure also shows that during the reproduction, the left tool reached an area on the lower-left of the pegboard that was not trained during the off-line stage. This is due to a failure during a ring transfer in which the ring fells out of the pegboard. As a consequence, the camera moved to an area further in the left than the trained behavior, which allowed it to keep the instruments within the camera view.

Regarding the vertical tracking, Figure 9b shows that there were two behaviors that are clearly identifiable: the zoom-out gesture described in the previous section, in which the camera got to the highest z positions, and the tracking in the z-axis during the normal performance of the task. Figure 10 shows examples of the zoom-out gesture during the training (Figure 10a) and the reproduction (Figure 10b) of the task. In both figures, a time window of the experiments are represented, and the particular instants of time in which the zoom-out gesture occurs are marked with shaded areas. At these instants, it can be appreciated how both tools raise in the z-axis, and the camera makes an outwards motion in response. Comparing the behavior during the training and the reproduction, it can be noted that during the training, the camera zooms out at a mean z position of 292 mm, while during the reproduction the camera gets a mean z value of 260 mm, around 30 mm less than the trained behavior. This has to do with the motion of the tools during both experiments. Analyzing the data, during the training stage, the tools reached a higher position than during the reproduction, which is the reason why the camera had a lower position in the latter case. However, Figure 10 shows how the zoom-out gesture is correctly detected by the system, and the trained behavior is performed.

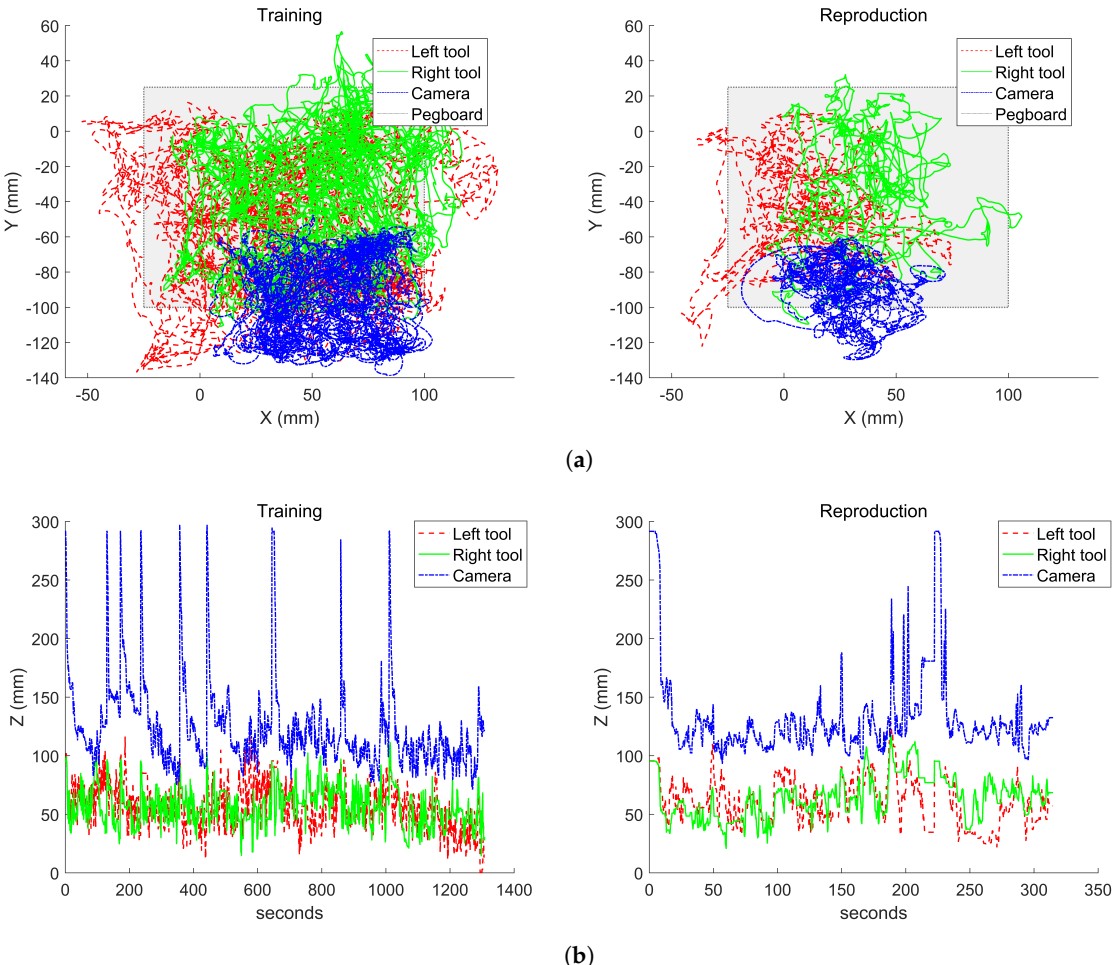

(**a**)

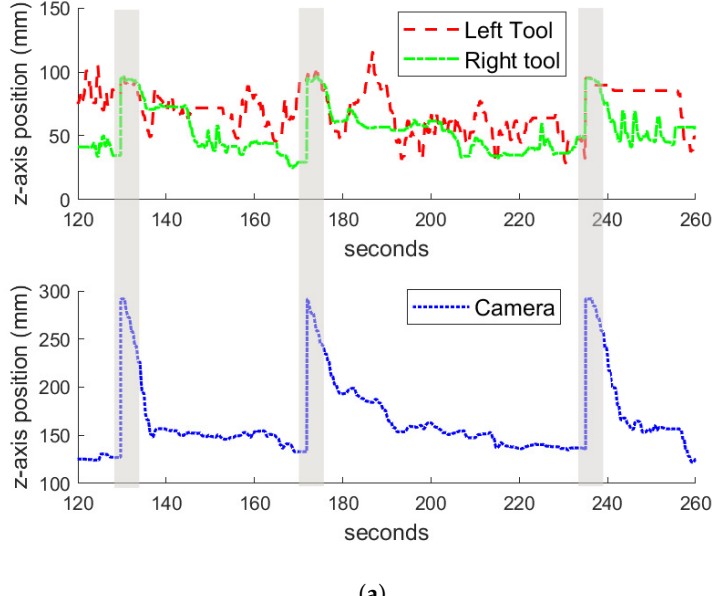

(**b**)

**Figure 9.** Comparison between the training and the reproduction experiments for the instruments tracking behavior: (**a**) tracking of the instruments in the horizontal plane, and (**b**) tracking in the vertical plane.

(**a**)

**Figure 10.** *Cont.*

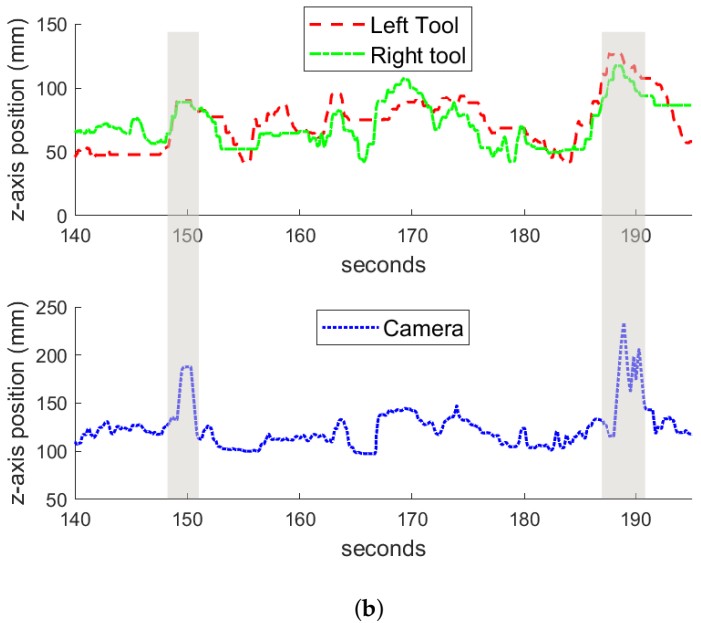

(**b**)

**Figure 10.** Zoom-out gesture during the (**a**) training and the (**b**) reproduction experiments. The shadow areas represent the moment in time in which the gesture occurs.

## 4. Discussion

This paper shows the feasibility of using learning from demonstration for autonomous camera guidance in laparoscopic surgery. Peg transferring was selected as a suitable task that demonstrated the surgeon skills. Around 43 min were spent to generate enough information to train a GMM with 20 Gaussians, which was considered enough to perform camera guidance with a 125 Hz real-time system. During the training stage, two behaviors were considered: instrument tracking and zoom-out. They were evaluated by repeating the same task for a period of 10 min. Both the attached video and figures, show that the previously defined behaviors were accomplished during the task reproduction stage, demonstrating that camera guidance is a suitable task to be carried out autonomously. However, there are several issues that remain to be investigated. The first one is related to the global frame pose for the task. In the experimental scenario, a well known global reference was used, i.e., the top left cylindrical rubber ring. However, in real surgery, the global reference frame would be difficult to choose, as it must be something fixed inside the abdominal cavity, and also, the position and dimension of the organs are different depending on the patient. The second issue is related to the task recognition. As stated in Section 1, there is a lot of work related to the detection of the surgical procedure stage in order to fix some parameters. It would be desirable to use these methods to detect the task that is being carried out by the surgeon, and therefore, apply the correct camera guidance model. Finally, the autonomous camera guidance method proposed in this work should be tested in a more complex task performed by expert surgeons. Moreover, in order to include it in a real surgery, the system must include a supervisor system that guarantees the patient's safety. The solution to these issues is proposed as future works.

**Supplementary Materials:** The following are available online at http://www.mdpi.com/2079-9292/8/2/224/s1, Video S1: Representational part of the task reproduction.

**Author Contributions:** Conceptualization, I.R.-B. and C.J.P.-d.-P.; methodology, C.J.P.-d.-P.; software, C.J.P.-d.-P., E.B.; validation, I.R.-B., C.L.-C.; writing—original draft, I.R.B., C.J.P.-d.-P.; funding acquisition, V.F.M.

**Funding:** This research was funded by the Spanish Ministry of Economy and Competitiveness under the grant number DPI2016-80391-C3-R.

**Conflicts of Interest:** The authors declare no conflict of interest.

**Abbreviations**

The following abbreviations are used in this manuscript:

| | |
|---|---|
| LfD | Learning from demonstration |
| GMM | Gaussian mixture model |
| EM | Expectation-maximization |
| BIC | Bayesian information criterion |
| GMR | Gaussian mixture regression |
| HMI | Human–machine interface |
| SW/HW | Software/Hardware |

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
