# Peer review of "Transferring Know-How for an Autonomous Camera Robotic Assistant"

_electronics, doi:10.3390/electronics8020224_

Round 1

Reviewer 1 Report

I am glad to review the manuscript. The paper aims at transferring know-how for an autonomous camera robotic assistant. It proposes a autonomous camera guidance approach for laparoscopic surgery. It is based on Learning from Demonstration (LfD). A robot observes how a human performs a task (i.e., the demonstration) and then it autonomously reproduces the human behavior to complete the same task (i.e., the imitation). The proposed approach has been validated using an experimental surgical robotic platform to perform peg transferring, a typical task that is used to train human skills in laparoscopic surgery. The paper is written clearly and is easy to follow. Here are some questions/comments that need to be answered/corrected:

1.     The last sentence of page 1: “Autonomous camera guidance would simplify the surgery and would reduce the operation time.”  needs reference or proof. Even if it was shown that it would in the laboratory setup and experiments, it cannot be concluded that it would simplify the surgery in real surgeries, or would reduce the operation time.

2.     Does master console in figures 2 and 3 show a simulated environment or is it camera vision? Does the operator have a pair of 3D glasses? Is the view 3D? Does the operator have a direct view of the robotic arms? These are not clear in the figures.

3.     What is the role of haptic devices and haptic feedback in this paper? Does it make any difference if a haptic device or a regular joystick be used to control the robots, as just the position information of the surgical tools are used in controlling the camera.

4.     How did the k=327.043 tuples calculated on line 201?

5.     Figure 6 a and b are not aligned horizontally. 

6.     In Figure 9a, the legends on the left and right panels are not symmetrical.

7.     How does a “good camera view” is defined and quantified in line 243 on page 10? 

8.     How does the reliability of the control system for the camera placement can be evaluated? How unwanted motions of the autonomous camera system are defined and avoided? 

9.     Page 2, Line 96, (P2x, P12y, P12z) should be changed to (P2x, P2y, P2z).

10.  Equation 7 (sigma pc/n) should be corrected.

11.  How should (Pi)n (the weight of the Gaussian) be defined? 

12.  Can system works well while being used by other surgeons? (trained by one surgeon and used by others)

13.  How preference of the surgeon is considered in training of the system? Do all surgeons move the camera in the same way or are there differences in the way they use the camera? Are these differences considered in the training? If not (which is the case for this paper), how can we conclude that the system will perform well if used by other surgeons?

Author Response

Dear reviewer and Managing Editor:

the point-by-point review to the comments of Reviewer 1 is attached as a PDF file. 

Reviewer 2 Report

The paper describes employing Gaussian probabilistic models to carry out autonomous camera placement for laparoscopic surgery.  A model is trained using learning-by-demonstration, and the model is then tested with the camera in an autonomous mode.

The paper is one of the best I have read in a while so I commend you on that. It is generally well written, the layout is excellent and clear, and your introduction gives a good overview of other methods and why your work is justified. Your conclusions are justified by your results and you give good limitations and follow up work. 

-- Major comments

-It is not clear whether the training and testing are both carried out with the exact same task, or if peg locations are randomised in training. This isn't very clear, and is crucial in showing generalisation of the method used. 

- There is no mention of safety considerations as far as I can see, which is obviously important when using autonomous systems in surgery. 

- The N value for the GMM is chosen to maintain real-time constraints, but these constraints aren't specified or justified.

- It would be interesting to see some comments from a surgeon on their experience with using the automated camera system. 

-- Minor comments

- line 23: "robot" replaces "robots"

- line 25: "However" replaces "But" (sentences should not start with "but")

- line 122: missing description of either mu_n or Sigma_n, not sure which

- equation (4): the EM algorithm is not defined or explained

- equation (5): The -L() function is not defined

- line 159: "On the other hand" can be removed (incorrect usage)

- line 176: expand "SW/HW" to "software/hardware"

- line 179: please reference the (NI-PXI) real-time hardware

- line 221: I think "right" should be "left"

- line 231: change "bad" to "poorly"

- general diagrams: Line clarity can be improved, I find it hard to see on some of them (be aware of colour blind limitations or if viewing in black & white)

Author Response

Dear reviewer and Managing Editor:

the point-by-point review to the comments of Reviewer 2 is attached as a PDF file.
